# Testing Proximal Optical Sensors on Quinoa Growth and Development

**Jorge Alvar-Beltrán [1],*** , **Carolina Fabbri [1]**, **Leonardo Verdi [1]**, **Stefania Truschi [2]**, **Anna Dalla Marta [1]** and **Simone Orlandini [1]**

[1] Department of Agriculture, Food, Environment and Forestry (DAGRI), University of Florence, 50144 Florence, Italy; carolina.fabbri@unifi.it (C.F.); leonardo.verdi@unifi.it (L.V.); anna.dallamarta@unifi.it (A.D.M.); simone.orlandini@unifi.it (S.O.)

[2] Fondazione per il Clima e la Sostenibilità, 50145 Florence, Italy; stefania.truschi92@gmail.com

\* Correspondence: jorge.alvar@unifi.it

**Abstract:** Proximal optical sensors (POSs) are effective devices for monitoring the development of crops and the nitrogen (N) status of plants. POSs are both useful and necessary in facilitating the reduction of N losses into the environment and in attaining higher nitrogen use efficiency (NUE). To date, no comparison of these instruments has been made on quinoa. A field experiment conducted in Tuscany, Italy, with different POSs, has assessed the development of quinoa with respect to N status. Three sets of POSs were used (SPAD-502, GreenSeeker, and Canopeo App.) to monitor quinoa development and growth under different types of fertilizers (digestate and urea) and levels of N fertilization (100, 50, and 0 kg N ha$^{-1}$). The present findings showed that in-season predictions of crop biomass at harvest by SPAD-502 and GreenSeeker optical sensors were successful in terms of the coefficient of determination ($R^2$ = 0.68 and 0.82, respectively) and statistical significance ($p < 0.05$), while the Canopeo App. was suitable for monitoring the plant´s canopy expansion and senescence. The relative error (RE%) showed a remarkably high performance between observed and predicted values, 5.80% and 4.12% for GreenSeeker and SPAD-502, respectively. Overall, the POSs were effective devices for monitoring quinoa development during the growing season and for predicting dry biomass at harvest. However, abiotic stresses (e.g., heat-stress conditions at flowering) were shown to reduce POSs' accuracy when estimating seed yields at harvest, and this problem will likely be overcome by advancing the sowing date.

**Keywords:** nitrogen fertilization; normalized difference vegetation index; yield and biomass prediction; abiotic stresses

## 1. Introduction

Given the well-documented nutritional properties of quinoa (*Chenopodium quinoa* Willd.), this crop has become an important food source worldwide [1]. To satisfy the current demand for quinoa, farmers are intensifying the use of conventional agricultural strategies. These strategies are sometimes not in line with sustainable agricultural practices, where farmers are increasingly encouraged to manage farming inputs correctly [2,3]. Besides reducing N losses into the environment, appropriate N management can increase both nitrogen use efficiency (NUE) and the economic profit of farmers [4,5]. Optimal N management (high yield production by minimizing N losses) can be achieved by adopting precision farming techniques capable of detecting N needs during the growing season [6]. A suitable approach is to adjust in-season N rates based on the prediction of potential yield [7]. To meet such demand, many optical monitoring devices have been developed in recent years [8–11]. Proximal optical sensors (POSs) base their readings on the detection of chlorophyll, an indicator of N content [12,13].

The use of chlorophyll meters has facilitated fertilization decision-making processes by adjusting N fertilization during the growing season, thereby increasing the seed N content and the final yield [14–17]. A well-known chlorophyll meter is the SPAD-502, a non-destructive handheld sensor that provides numerical estimates of the chlorophyll content in a leaf. The SPAD-502 clips onto a single leaf and records light intensity ranging between 650 and 940 nm by measuring the absorbance of light emitted by the leaf [18]. However, chlorophyll meters can only sample a relatively small area (SPAD-502 6 $mm^2$ and MULTIPLEX 1250 $mm^2$) and therefore multiple repetitions are required to obtain a representative measurement [19].

Regarding reflectance sensors, ground-based active crop canopy sensors have been developed for site-specific N management in cereals [12,20]. Crop reflectance sensors are also promising methods for monitoring the N status of vegetable crops. The main advantage of these sensors is that an individual observation can gather information over a much larger surface than that measured by chlorophyll leaf meters [19]. GreenSeeker (NTech Industries Inc., Ukiah, CA, USA) is a widely used active crop sensor. This crop sensor measures radiation in the red (660 nm) and near infrared (770 nm) wavelength spectrum, providing information on the Normalized Difference Vegetation Index (NDVI), an indicator of the N content and seed yield potential [7,21].

Up until now, multiple studies have used POSs to monitor N content, as well as to predict seed yields of winter wheat, observing a correlation between NDVI and N uptake with seed yield at harvest [22–30]. For instance, Inman et al. [24] found that NDVI was useful for estimating grain yields of irrigated maize, while canopy cover readings, at the late-filling stage, were useful for predicting winter wheat yields. Other studies demonstrated that NDVI could predict yield, determine crop N requirements, and assist N use efficiency improvements in wheat and corn [31,32]. Molin et al. [33] reported a significant relationship between N rates and NDVI with GreenSeeker. Canopy cover sensors also showed a high performance in predicting sugarcane yields (using diverse canopy proximal sensors, SPAD-502 and Crop Circle) [34–36]. Interestingly, a strong relationship ($R^2 = 0.77$) between NDVI and seed yield was found in the V8 leaf-stage of maize, but not in the other leaf stages [31]. Meanwhile, some studies affirm that the advantage of using an in-season indicator for maize, with reflectance obtained at V6-V7, is that the plant behaves as an integrator of conditions and stresses already experienced at early growing stages [37].

Moreover, Xue et al. [38] noted that N uptake and rice yields were positively correlated with NDVI at the tillering and panicle formation phase. For Chlorophyll meter measurements, Bavec and Bavec [39] found that Zadok Growth Stages-ZGS 45–50 were correlated with winter wheat seed yields (r = 0.54, $p < 0.01$). In addition, Vidal [40] detected, using SPAD, that around 80% of the variation in wheat seed yield was observed at ZGS 45–69. For the Canopeo App., a tool for measuring the fractional green canopy cover of plants, a strong relationship between Canopeo images and light quantum sensors was observed when monitoring the canopy cover of soybean [41]. Additionally, Chung et al. [42] embraced an innovative approach with the Canopeo App.; by taking vertical images of the crop, they observed a strong correlation between plant height in-season measurements and biomass production.

Until now, most of the spectral studies have evaluated cereal crops and, to a lesser extent, vegetables. However, the number of studies testing different POSs on quinoa development and growth are scarce. Any advances on testing POSs on quinoa development and growth as well as on the ability of seed yield in-season prediction will benefit farmers, helping them improve decision-making related to N fertilization. Therefore, the adequate management of N in quinoa production is imperative given the countless services that it renders to farmers and to the environment.

The objective of this study was to evaluate the effectiveness of multiple POSs for tracing the N requirements during the growing season. In order to monitor the crop dynamics, to assess the N use, as well as to determine the optimal time for predicting yields and biomass, the following set of optical instruments were selected: SPAD-502, GreenSeeker, and Canopeo App. This study's comparison of multiple and recently developed POSs (e.g., Canopeo App.) and their applications in

crop growth and the development of quinoa, makes it unique and different from the others in the literature. Finally, an inter-comparison assessment of the POSs was performed to test the accuracy of the devices throughout the experiment, which was designed using both types of fertilizer (urea and digestate) and N fertilization rates (100, 50, and 0 kg N ha$^{-1}$).

## 2. Materials and Methods

### 2.1. Experimental Design

This research was conducted at the experimental field of the Istituto Tecnico Agrario in Florence, Italy, (WGS84: 43°47′06″ N; 11°13′06″ E; 40 m.a.s.l.) between May and August 2019 (Figure 1). The study was organized in a completely randomized design with one factor and five N fertilization levels, each with three replicates. The N fertilization levels were as follows: urea $CO(NH_2)_2$ at a rate of 100 and 50 kg N ha$^{-1}$, digestate at a rate of 100 and 50 kg N ha$^{-1}$, and a control with 0 kg N ha$^{-1}$ (hereafter: 100-urea, 50-urea, 100-digestate, 50-digestate, and 0-control). Fertilizers (urea and digestate) were applied 28 and 49 days after sowing (DAS) and incorporated into the soil through mechanical tillage (harrowing at 20 cm depth). Experimental plots of 2.10 × 1.80 m (3.78 m$^2$) were subdivided into four rows, spaced 70 cm apart and with 10 cm between plants. Quinoa was sown on 8 May 2019, at a sowing rate of about 140–150 plants m$^{-2}$ and harvested on 27 August 2019 (110 DAS). The selected genotype was Titicaca, a hybrid developed at the University of Copenhagen, Denmark.

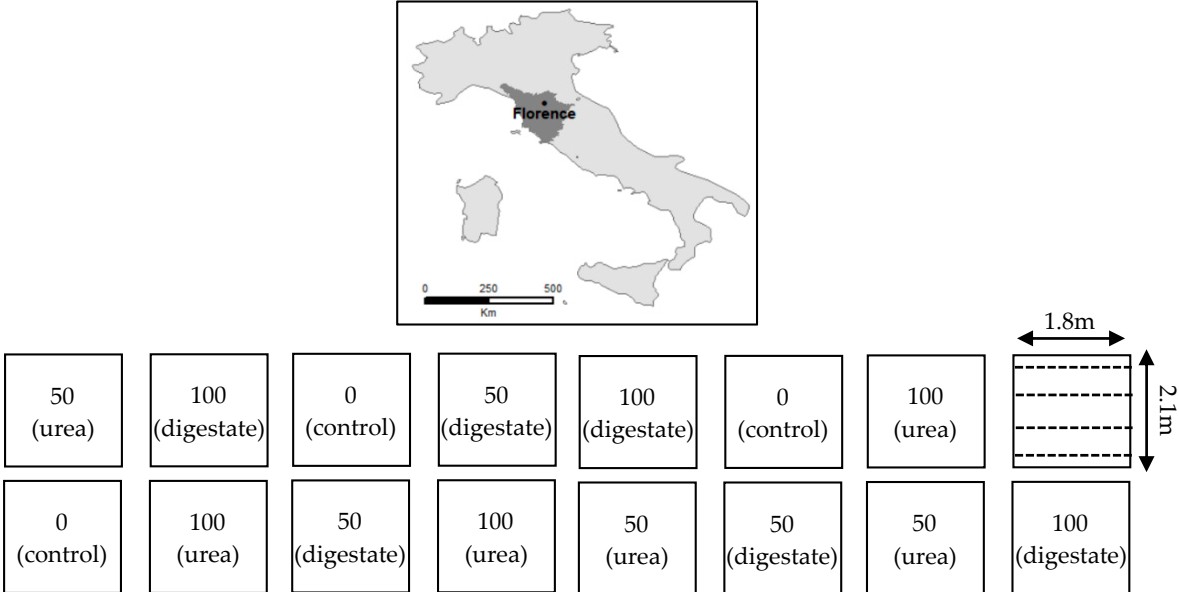

**Figure 1.** Location of the experimental site and research design with different N fertilization levels (100/50 kg N ha$^{-1}$ urea, 100/50 kg N ha$^{-1}$ digestate, and 0 kg N ha$^{-1}$ -control).

### 2.2. Plant Sampling and Weather Measurements

The measured growth staging phases of quinoa using the BBCH codes [43] was as follows: BBCH-10 (cotyledons fully emerged), BBCH-11 (first pair of leaves visible), BBCH-14 (four pairs of leaves visible), BBCH-51 (leaves surrounding inflorescence separated), BBCH-60 (beginning of anthesis), BBCH-81 (milky grain), and BBCH-93 (leaves of the first half portion of the plant are dead) at 7, 16, 28, 33, 49, 77, and 110 DAS, respectively (76, 189, 377, 482, 833, 1499, and 2263 cumulative growing degree days CGDD, respectively, Equation (1)). Plant height (PH) was measured at 30, 49, 63, and 110 DAS (414, 833, 1184, 2263 CGDD, respectively) for 10 randomly selected plants (Table 1). The number of branches per plant (BP in number), the length and width of the panicle (LP and WP in cm and mm, respectively), and stem diameter (SD in mm) were measured for 5 plants. At harvest, the seed yield per plant (SYP in g plant$^{-1}$, including all plants from the middle rows) and thousand-seed

weight (TSW in g, 3 samples per plot) were measured. All the information was collected from the central rows to avoid any side effects. The daily maximum, minimum, and mean air temperatures (°C), as well as precipitation (mm), were recorded by a weather station located at 0.5 km from the experimental field.

$$CGDD = \frac{(Tmax + Tmin)}{2} - Tbase \tag{1}$$

where Tmax and Tmin correspond to the daily maximum and minimum temperature, while Tbase (3 °C) corresponds to the base temperature of quinoa [44].

**Table 1.** Flowchart of agronomic activities as days after sowing (DAS) and cumulative growing degree days (CGDD), parameters measured, and number of samples.

| Measurement Date (DAS) | CGDD | Agronomic Practices | Parameter(s), Devices and Number (#) of Samples Per Plot |
|---|---|---|---|
| Before sowing | - | - | Soil samples (#3) |
| 0 | - | Sowing date | - |
| 7 | 76 | - | BBCH-10 (#38) |
| 16 | 189 | - | BBCH-11 (#38) |
| 28 | 377 | 1st fertilization Weeding | BBCH-14 (#38) |
| 30 | 414 | - | PH; NDVI and Canopy cover (#5); Chlorophyll (#25) |
| 33 | 482 | - | BBCH-51 (#38) |
| 49 | 833 | 2nd fertilization Weeding | BBCH-60 (#38); PH; NDVI; Canopy cover (#5); Chlorophyll (#25) |
| 55 | 991 | - | Canopy cover (#5) |
| 63 | 1184 | - | PH (#5) |
| 70 | 1330 | - | NDVI and Canopy cover (#5); Chlorophyll (#25) |
| 77 | 1499 | - | BBCH-81 (#38) |
| 84 | 1658 | - | NDVI and Canopy cover (#5); Chlorophyll (#25) |
| 93 | 1870 | - | NDVI and Canopy cover (#5); Chlorophyll (#25) |
| 110 | 2263 | Harvest | BBCH-93 (#38); PH, BP, LP, WP, SD (#5) |
| After harvesting | | - | SYP (#38); TSW (#38) |

Spectral characteristics of leaves and biomass of quinoa plants were measured using the GreenSeeker (NDVI), SPAD-502 (chlorophyll), and Canopeo App. (canopy cover) at 30, 49, 70, 84, and 93 DAS (414, 833, 1330, 1658, 1870 CGDD, respectively), including 5 plant readings per plot, except for SPAD-502 (average of 5 readings per plant for 5 plants per plot). In order to obtain a better estimation of the onset of leaf senescence, an additional observation of the canopy cover was recorded with the Canopeo App. at 55 DAS (991 CGDD) (Table 1).

*2.3. Proximal Optical Sensing*

The POSs were used to monitor and evaluate the N content in the plant during the growing season. The POSs calculated the light absorbed by the leaf, which was proportional to the chlorophyll content in the leaf. The GreenSeeker hand-held sensor (NTech Industries Inc., Ukiah, CA, USA) was used to determine spectral reflectance from the canopy expressed as NDVI. The device was equipped with a self-illumination system with both red (660 nm) and near infrared (780 nm) wavelengths. The device was held vertically, approximately 60 cm above the canopy cover (to avoid bare soil reflectance), and measurements were taken from the middle rows. The SPAD-502 meter (Konica Minolta, Inc., Tokyo, Japan) was used to detect the relative chlorophyll content by measuring light absorbance between 650 nm (red) and 940 nm (NIR). The units provided by the SPAD-502 meter were dimensionless and were directly related to the chlorophyll content. The Canopeo App. was programmed by Matlab (Mathworks, Inc., Natick, MA, USA) integrating colored images (red, green, and blue (RGB)) for monitoring the surface area covered by crops. Green canopy was displayed in white pixels, while bare soil was presented in black pixels. The Canopeo App. results were reported as a percentage of the green canopy cover and measurements were taken using a smartphone held 60 cm above the green canopy cover. The SPAD-502 was calibrated whenever the meter was switched on, while GreenSeeker and Canopeo App. did not require a calibration prior to start measuring.

## 2.4. Data Analysis

The findings of the present study were analyzed using the Minitab 18 statistical software. The ANOVA and post-hoc Fisher tests were used to explore the possible pairwise comparisons of means and the main factor effect of fertilizer type (digestate and urea) and N fertilization rates (100, 50, and 0 kg N ha$^{-1}$). In addition, the coefficient of determination ($R^2$, Equation (2)) was used to examine the proportion of the variance on the dependent variable (biomass) that was predictable from the independent variable (POSs). The correlation significance was verified using T tests ($p < 0.05$). To estimate the sensitivity of the POSs, a relationship between observed readings was calculated via linear regression. The relative error (RE%, Equation (3)) and root mean square error (RMSE, Equation (4)) were determined and the prediction was considered excellent if RE < 10%, good if 10–20%, fair if 20–30% and poor if >30%, respectively [45,46].

$$R^2 = \frac{\sum (yest - \hat{y})^2}{\sum (y - \hat{y})^2} \tag{2}$$

where *yest* is the estimated value, $\hat{y}$ is the mean of observed data, and *y* is the observed value.

$$RE\% = \frac{(P_i - O_i)^2}{O_i} \times 100 \tag{3}$$

$$RMSE = \sqrt{\sum_{i=1}^{n} \frac{(P_i - O_i)^2}{n}} \tag{4}$$

where $P_i$ is the measured value and $O_i$ is the actual value.

## 3. Results

### 3.1. Soil and Weather Conditions

The soil texture of the present experiment was characterized as sandy loam (0–10 cm depth) (Table 2). The concentration of organic carbon was 1.94% and the N content 0.15% (1.5 mg N kg$^{-1}$). The C:N ratio was 12.9, resulting in a longer time for organic matter to decompose.

**Table 2.** Physical–chemical properties of the soil prior to sowing at 0–10 cm depth.

| Parameter | Unit | Characteristics |
|-----------|------|-----------------|
| Sand | % | 56 |
| Silt | % | 29 |
| Clay | % | 15 |
| Texture | | Sandy-Loam |
| C | % | 1.94 |
| N | % | 0.15 |
| C/N | | 12.9 |

The mean temperature recorded during the growing season was 23.5 °C (Figure 2). This value was in line with the optimal growing conditions for quinoa (10–25 °C). Extreme temperatures were observed at flowering (starting 49 DAS), with temperatures exceeding 36 °C during an entire week. The total amount of precipitation recorded during the growing season was 160 mm, of which the majority (104 mm) occurred during the first 20 days. Optimal water conditions were reported during the vegetative stage, whereas long dry periods, except for occasional showers, characteristic in July and August, coincided with the flowering and seed-filling stages. As a result of the heat and drought stress conditions during flowering and seed-filling phases, losses in yield were reported at harvest.

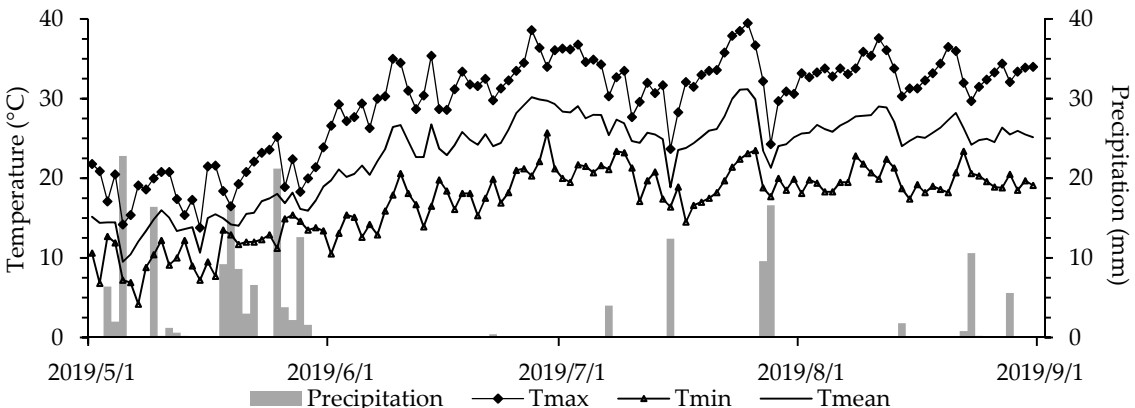

**Figure 2.** Meteorological data observed during the growing season.

### 3.2. Proximal Optical Sensor Readings

The SPAD-502 and GreenSeeker data on biomass reported a similar rate of N content increase during the vegetative stage up to flowering (49 DAS) (Figure 3). GreenSeeker readings for different N levels were statistically different ($p < 0.05$), particularly after the second fertilization event (49 DAS). At 70 DAS, NDVI GreenSeeker records were higher under 100-urea/digestate (0.48 and 0.47, respectively) than for 50-urea/digestate (0.44 and 0.40, respectively) and the 0-control (0.40). Leaf SPAD-502 measurements displayed a peak in N content at 49 DAS (average value of 50), whereas the GreenSeeker showed the highest NDVI values at 70 DAS (0.44 average for all treatments). The Canopeo App. readings (Figure 4) recorded the maximum canopy cover (about 65%) at flowering (49 DAS), with no differences between treatments. Overall, the rate of leaf senescence was more rapid under non-fertilized treatments (control). From 49 to 70 DAS, the rate of canopy cover loss was twice as fast in the control (46%) than under N fertilized treatments (28%). At 84 DAS, all treatments reached the lowest canopy cover (less than 14%), remaining constant until the physiological maturity of the plant.

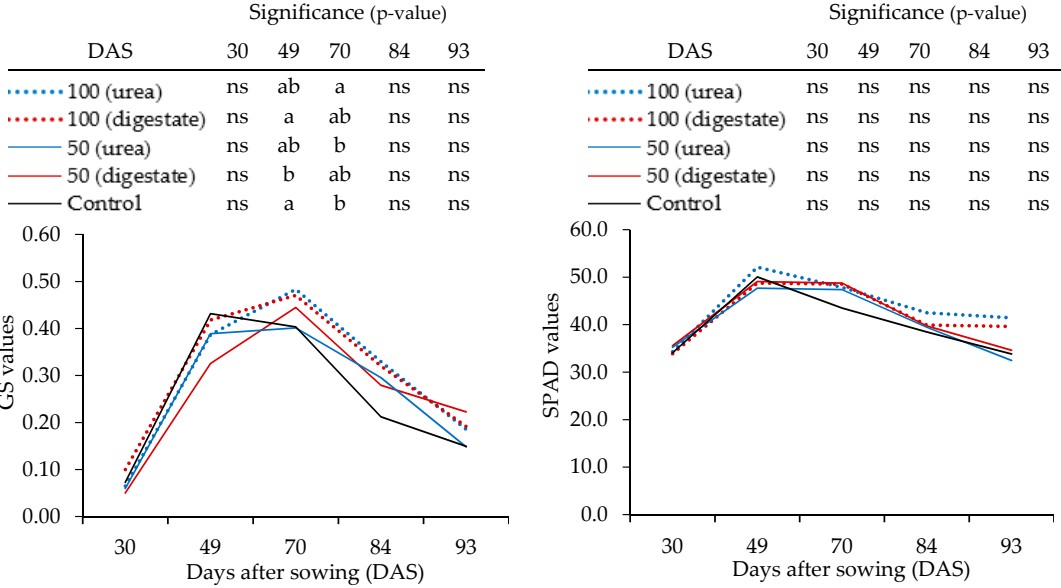

Legend: units for GreenSeeker (GS) and SPAD-502 values are dimensionless

Legend: means that do not share a letter were significantly different (a, b, c) at the 5% probability level using the Fisher test and ns (not significant)

**Figure 3.** Time series of GreenSeeker-GS (left) and SPAD-502 readings (right) for different N fertilization treatments (100/50 kg N ha$^{-1}$ urea, 100/50 kg N ha$^{-1}$ digestate, and 0 kg N ha$^{-1}$ -control).

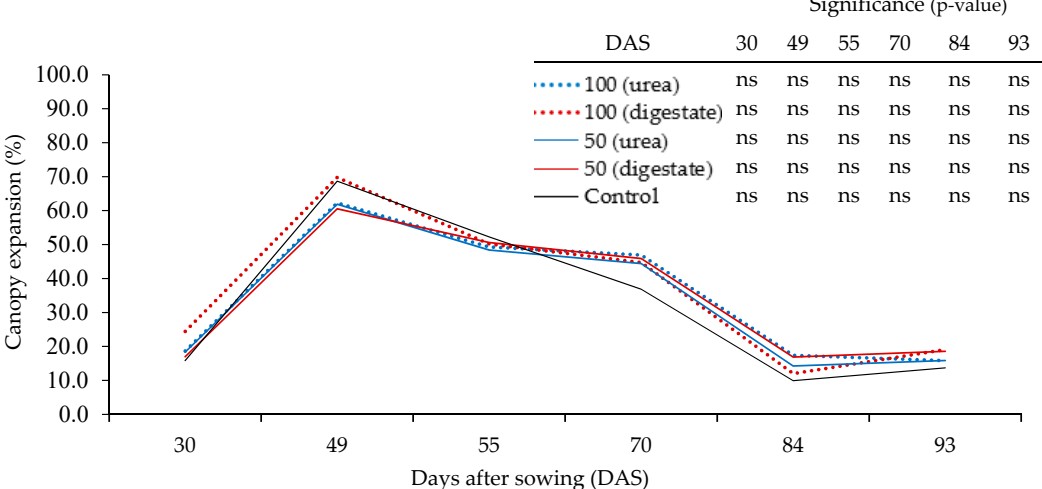

Legend: means that do not share a letter were significantly different (a, b, c) at the 5 % probability level using the Fisher test and ns (not significant)

**Figure 4.** Canopy cover expansion (%) for different N fertilization treatments (100/50 kg N ha$^{-1}$ urea, 100/50 kg N ha$^{-1}$ digestate, and 0 kg N ha$^{-1}$ -control).

A high coefficient of determination ($R^2$) showed an agreement between POSs and crop development parameters (Figure 5 and Table 3). Significant differences were observed when examining the relationship between SPAD-502 and GreenSeeker ($R^2 = 0.81$, $p < 0.05$ average of all values), and between SPAD-502 and Canopeo App. ($R^2 = 0.78$, $p < 0.05$ average of all values). However, no significant differences were reported for the interaction between GreenSeeker and Canopeo App. ($R^2 = 0.52$, $p > 0.05$ average of all values). The control showed the highest $R^2$ values between POSs (GreenSeeker vs. Canopeo App., SPAD-502 vs. Canopeo App. and SPAD-502 vs. GreenSeeker, with a $p < 0.05$, $p < 0.01$, and $p < 0.01$, respectively). Instead, the lowest $R^2$ values were observed under high N fertilization rates (both under 100-urea/digestate).

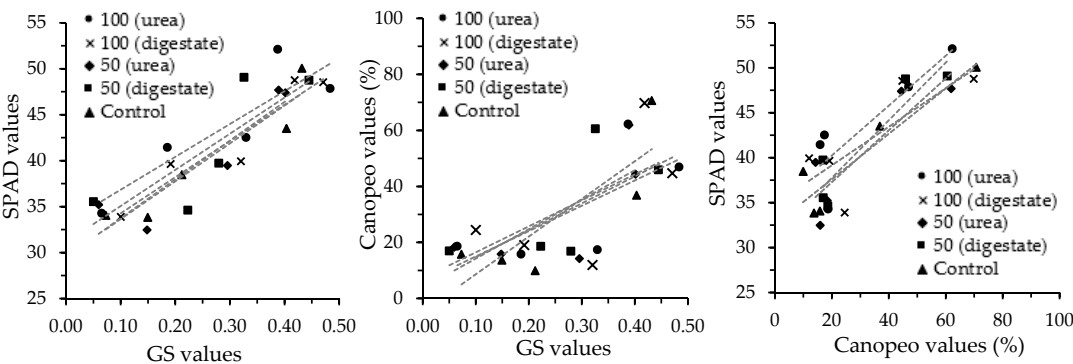

Legend: each treatment value corresponds to the average of three replicates

Legend: units for GreenSeeker (GS) and SPAD-502 values are dimensionless

**Figure 5.** Relationship between different POSs (SPAD-502, GreenSeeker, and Canopeo App.) and N fertilization treatments (100/50 kg N ha$^{-1}$ urea, 100/50 kg N ha$^{-1}$ digestate, and 0 kg N ha$^{-1}$ -control).

The Pearson correlation analysis was performed to identify which POSs predicted the best biomass and seed yields at harvest. In terms of biomass, results showed that both SPAD-502 observations at 84 DAS ($R^2 = 0.68$, $p < 0.05$) and GreenSeeker measurements at 70 DAS ($R^2 = 0.82$, $p < 0.05$) were highly accurate at predicting biomass at harvest (Figure 6). The highest $R^2$ values were displayed by the GreenSeeker at 70 DAS, displaying a strong correlation between the device readings and biomass at harvest. The regression analysis also showed a strong relationship between GreenSeeker and SPAD-502

with biomass at harvest (independent and dependent variables, respectively). As a result of a high accuracy between the POSs and biomass at harvest, the model was validated for in-season predictions. However, the POSs were unable ($p > 0.05$) to predict biomass at harvest during early growing stages. Another approach (bias method) was used to assess the performance of the SPAD-502 and GreenSeeker models at 70 and 84 DAS. The results showed a positive performance with values close to 0 for both devices (0.60 and 0.95, respectively). Hence, the performance of both devices and models was corroborated, with the lowest values being observed when using the SPAD-502 at 84 DAS. In addition, the RMSE displayed lower values for the SPAD-502 model when compared to the GreenSeeker model. The RE% results for biomass predictions were 5.80% and 4.12% for the GreenSeeker and SPAD-502, respectively. In both cases, these prediction values were considered excellent according to the RE% criterion developed by [45].

**Table 3.** Lineal regression, coefficient of determination ($R^2$), and statistical hypothesis testing (*p*-value) for the interaction between different POSs (SPAD-502, GreenSeeker, and Canopeo App.) and N fertilization treatments (100/50 kg N ha$^{-1}$ urea, 100/50 kg N ha$^{-1}$ digestate, and 0 kg N ha$^{-1}$ -control).

| | SPAD vs. GreenSeeker | | | Canopeo vs. GreenSeeker | | | SPAD vs. Canopeo | | |
|---|---|---|---|---|---|---|---|---|---|
| Treatment | Reg. Line | $R^2$ | Sig. | Reg. Line | $R^2$ | Sig. | Reg. Line | $R^2$ | Sig. |
| 100 (urea) | y = 35.8x + 33.3 | 0.77 | * | y = 90.5x + 6.0 | 0.50 | ns | y = 0.28x + 34.7 | 0.75 | * |
| 100 (digestate) | y = 41.2x + 29.5 | 0.90 | ** | y = 96.0x + 5.2 | 0.40 | ns | y = 0.21x + 34.8 | 0.62 | ns |
| 50 (urea) | y = 42.2x + 29.5 | 0.84 | ** | y = 105.1x + 3.8 | 0.56 | ns | y = 0.29x + 31.5 | 0.78 | * |
| 50 (digestate) | y = 39.3x + 31.1 | 0.66 | * | y = 92.8x + 7.2 | 0.44 | ns | y = 0.32x + 31.4 | 0.86 | ** |
| Control | y = 41.2x + 29.5 | 0.89 | ** | y = 134.5x − 4.8 | 0.70 | * | y = 0.25x + 32.6 | 0.87 | ** |

Legend: *** ($p < 0.001$), ** ($p < 0.01$), * ($p < 0.05$), ns ($p > 0.05$).

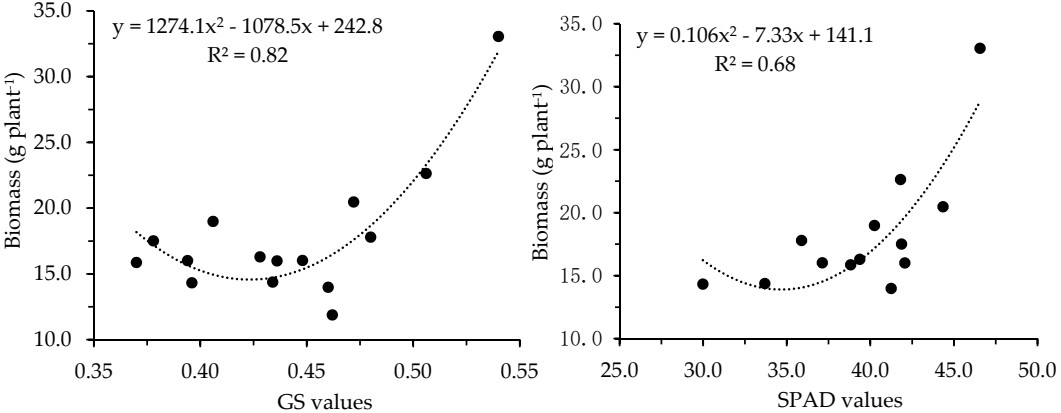

Legend: units for GreenSeeker (GS) and SPAD-502 values are dimensionless

**Figure 6.** Polynomial regression between SPAD-502 (at 70 DAS) and GreenSeeker-GS (at 84 DAS) observations with biomass at harvesting (at 110 DAS).

### 3.3. Effect of N Fertilization on Crop Growth and Development

In terms of crop growth attributes, quinoa performed better under urea fertilization than under digestate, but the difference was not significant. The highest seed yields per plant were obtained using 100-urea (4.96 g plant$^{-1}$), whereas the lowest yields were obtained using 50-digestate (3.75 g plant$^{-1}$) (Table 4). The plants were unaffected ($p > 0.05$) by different N fertilization rates in terms of height (96 cm average of all treatments) and showed a similar growth pace during the growing season (Figure 7). Additional crop parameters, such as number of branches, showed significant differences ($p < 0.05$) with increasing N fertilization rates (11.7 branches under 100-urea and 8.3 branches under 0-control). Instead, no significant differences ($p > 0.05$) were depicted between treatments for panicle length (21.3 cm in average) and kernel weight (1.33 g in average).

**Table 4.** ANOVA (± SD) and Fisher test for different crop parameters and N fertilization treatments (100/50 kg N ha$^{-1}$ urea, 100/50 kg N ha$^{-1}$ digestate, and 0 kg N ha$^{-1}$ -control).

| Fertilizer Type | N Rate (kg N ha$^{-1}$) | Yield (g Plant$^{-1}$) | Biomass (g Plant$^{-1}$) | 1000-Seed Weight (g) | Harvest Index (%) | Branches Plant (Number) | Panicle Length (cm) | Panicle Width (mm) | Stem Diameter (mm) |
|---|---|---|---|---|---|---|---|---|---|
| Digestate | 100 | 3.96 ± 0.69 | 19.16 ± 4.28 | 1.27 ± 0.15 | 20.9 ± 2.42 | 10.0 ± 2.42**ab** | 21.7 ± 2.44 | 43.5 ± 9.5 | 7.33 ± 1.21 |
| | 50 | 3.75 ± 0.77 | 15.94 ± 1.91 | 1.37 ± 0.20 | 23.6 ± 3.46 | 9.8 ± 1.73**ab** | 19.7 ± 2.53 | 61.3. ± 17.5 | 7.60 ± 1.83 |
| Urea | 100 | 4.47 ± 2.11 | 20.32 ± 11.22 | 1.39 ± 0.03 | 22.7 ± 2.22 | 11.7 ± 1.62**a** | 21.7 ± 1.50 | 56.0 ± 15.2 | 7.80 ± 1.83 |
| | 50 | 4.26 ± 0.53 | 17.05 ± 1.69 | 1.32 ± 0.09 | 25.0 ± 3.07 | 10.3 ± 2.04**ab** | 22.7 ± 1.30 | 49.9 ± 2.0 | 8.47 ± 1.10 |
| Control | 0 | 4.22 ± 0.63 | 15.94 ± 1.91 | 1.28 ± 0.00 | 22.4 ± 3.27 | 8.3 ± 0.50**b** | 20.9 ± 1.47 | 43.1 ± 6.91 | 6.93 ± 0.31 |
| | Digestate | 3.86 ± 0.66 | 17.55 ± 3.45 | 1.32 ± 0.17 | 22.3 ± 3.40 | 9.9 ± 1.89**ab** | 20.7 ± 2.50 | 52.4 ± 15.9 | 7.47 ± 1.40 |
| | Urea | 4.37 ± 1.38 | 18.69 ± 7.40 | 1.36 ± 0.07 | 23.9 ± 2.72 | 11.0 ± 1.82**a** | 22.2 ± 1.37 | 52.9 ± 10.3 | 8.13 ± 1.40 |
| | Control | 4.22 ± 0.63 | 15.94 ± 1.59 | 1.28 ± 0.00 | 22.4 ± 3.27 | 8.3 ± 0.50**b** | 20.9 ± 1.47 | 43.1 ± 6.91 | 6.93 ± 0.31 |
| | 100 | 4.22 ± 1.43 | 19.74 ± 7.62 | 1.33 ± 0.12 | 21.8 ± 2.30 | 10.8 ± 2.06 | 21.7 ± 1.81 | 49.8 ± 13.2 | 7.57 ± 1.41 |
| | 50 | 4.00 ± 0.65 | 16.49 ± 1.72 | 1.35 ± 0.14 | 24.3 ± 3.40 | 10.0 ± 1.71 | 21.2 ± 2.46 | 55.6 ± 12.8 | 8.03 ± 1.43 |
| | 0 | 4.22 ± 0.63 | 15.94 ± 1.59 | 1.28 ± 0.00 | 22.4 ± 3.27 | 8.3 ± 0.50 | 20.9 ± 1.47 | 43.1 ± 6.91 | 6.93 ± 0.31 |

Legend: means that do not share a letter were significantly different (a, b, c) at 5 % probability level using the Fisher test.

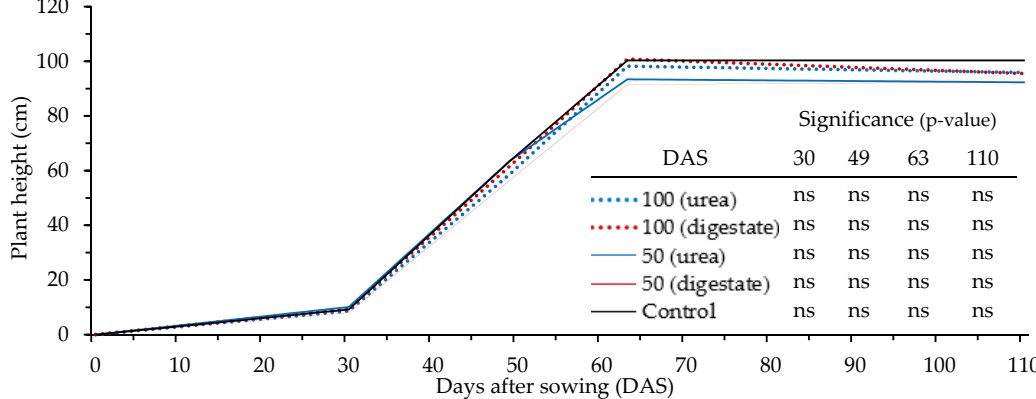

Legend: observations at 30, 49, 63, and 110 DAS; with a lineal interpolation for the remainder of the time series
Legend: means that do not share a letter were significantly different (a, b, c) at the 5 % probability level using the Fisher test and ns (not significant)

**Figure 7.** Evolution of plant height (cm) during the growing season for different N fertilization treatments (100/50 kg N ha$^{-1}$ urea, 100/50 kg N ha$^{-1}$ digestate, and 0 kg N ha$^{-1}$ -control).

In terms of crop development, significant differences ($p < 0.05$) between different levels of N fertilization were noted at the timing of fertilization (28 and 49 DAS) or right after. First, some differences were noted at BBCH-51 (at 33 DAS and 482 CGDD), where plants treated with N fertilizer (100/50-urea/digestate) developed slower than the control (0-control). Significant differences ($p < 0.05$) between fertilized treatments (100/50-urea/digestate) and the control (0-control) were also reported at BBCH-60 (at 49 DAS and 833 CGDD) and during BBCH-81 (at 77 DAS and 1499 CGDD). These differences were the result of a longer seed filling phase among fertilized plants than the control (non-fertilized).

## 4. Discussion

Even though some differences were observed in terms of crop growth and development between fertilizer types (digestate and urea) and N fertilization rates (100/50 kg N ha$^{-1}$ urea, 100/50 kg N ha$^{-1}$ digestate, and 0 kg N ha$^{-1}$-control), overall, quinoa was not very responsive to increasing N fertilization. As reported by Jacobsen [47], this could be explained by the low nutrient requirements of the crop, or by the high N content present in the soil, in the form of mineral nitrogen, before sowing (0.15% equivalent to 1500 mg kg$^{-1}$), resulting in insignificant differences between treatments, and, therefore, on POSs observations. There are many studies, particularly with cereal crops, underlying that optical sensors saturate or reach a plateau when there is no N limitation on crop performance [48–50].

In addition, this study has highlighted the potential use of POSs for the sustainable management of N during the growing season of quinoa. However, prior to the present results, comparative studies with different POSs had not yet been performed on quinoa. In addition, POSs are widely being applied on other cereals, such as wheat, maize, and rice [51–54]. These instruments are easy to manage and are effective for assessing the N status of plants. Optical indices are subject to variation, because light reflectance is influenced by changes in canopy structure and the N content of the plant during the growing season [55]. Hence, it is important to assess the characteristics of biomass to determine the relationship between optical indices and the N status of plants, and for predicting biomass and seed yield at harvest [56].

The present research results were in harmony with those reported previously [25,55]. SPAD-502 and GreenSeeker observations were found to be reliable when predicting biomass at harvest (Table 3), particularly if the observations were taken during the milky grain stage (70–84 DAS) [25,55]. In this experiment, no relationships were found between observations of the POSs during the early growth stages and biomass at harvest. This is attributable to the low expansion of the canopy cover and the high reflectance from the soil, which influenced the reading by the POSs [57]. However, once the

fertilization treatments were implemented, the readings were more reliable because each device was capable of discriminating N content and crop biomass from the bare soil, respectively. SPAD-502 values and biomass at harvest were lower in the control, with SPAD-502 values increasing at higher N fertilization rates. However, GreenSeeker could predict earlier in-season biomass at harvest when compared to the SPAD-502. This was explained by the fact that plant senescence played a crucial role on NDVI values both in terms of N content and vegetative expansion [58]. The optimal equation for the estimation of biomass is reported in Table 3, while the accuracy of the models was evaluated using several statistical error indices (e.g., RMSE and RE%).

The strong relationship reported between SPAD-502 and GreenSeeker readings during the growing season confirmed that both devices were precise at monitoring the N status of the plant (Table 5). However, the present study failed to predict in-season seed yields at harvest with the POSs, unlike previous studies [24,34,35]. This was invariably due to abiotic stresses (e.g., heat and water stress conditions) occurring during the key phenological phases (flowering and seed grain formation), which influenced the observations of the POSs, and consequently on the estimation of final yields [59]. Some authors reported similar interference, affirming that reflectance measurements and vegetation indices may be influenced by abiotic (e.g., water stress and irradiance) and biotic conditions (e.g., pest and diseases) [60,61]. In addition, abiotic stresses were shown to reduce seed weight, as well as overall seed production, with similar observations being made previously in Greece and in Burkina Faso [62,63]. The Canopeo App. was able to monitor a similar rate of leaf senescence for quinoa to that reported for winter wheat, with a more rapid leaf senescence occurring for the non-fertilized treatments [64]. The SPAD-502 readings reported in this study agreed with those of Alvar-Beltrán et al. [65] using the same genotype of quinoa, Titicaca. The latter study reported a chlorophyll content peak at 40 DAS, with an average SPAD-value of 46.9. In contrast, the highest values in the present study were shown to occur at 49 DAS, with an average SPAD-502 reading of 49.5.

**Table 5.** Observed advantages and disadvantages of different Proximal Optical Sensors (POSs) on monitoring quinoa growth and development.

| POSs | Advantages | Disadvantages |
|---|---|---|
| SPAD-502 | Precise on measuring N content evolution. Accurate on predicting in-season biomass at harvest. | Not accurate on predicting in-season yields at harvest under heat-stress conditions. Small sample area. |
| GreenSeeker | Precise on measuring N content evolution Accurate on predicting in-season biomass at harvest. | Not accurate on predicting in-season yields at harvest under heat-stress conditions. |
| Canopeo App. | Significant canopy areas can be measured Precise on monitoring canopy cover expansion. | Frequent weeding required to obtain accurate results. |

This research results corroborated those made in South Italy by Pulvento et al. [66], showing a considerable decline in Titicaca yields when sown in May rather than in April (from 3.3 to 1.5 t ha$^{-1}$, equivalent to 16.5 and 7.5 g plant$^{-1}$, respectively). Moreover, the present study reported a lower harvest index (40%) and kernel weight (2.1 g) to that reported by Pulvento et al. [66]. From an agro-climatic perspective, the central regions of Italy are generally more affected by heat-stress conditions during the summer months when compared to southernmost regions, where temperatures are generally milder due to the thermoregulatory effect of the sea.

## 5. Conclusions

The potential beneficiaries of this study are the increasing number of quinoa farmers that can improve decision-making related to N fertilization based on in-season seed yield prediction. In addition,

since a minor part of the N applied is recovered by crops, appropriate N management is essential for reducing losses to the environment and diminishing associated environmental problems.

The present research showed optical sensors to be both effective and suitable for monitoring the development of quinoa and predicting its in-season biomass at harvest. The use of these sensors for predicting in-season seed yield at harvest should be further explored in order to encompass more long-term records, as well to test them under differing weather conditions. The Canopeo App. was shown to be the better sensor in assessing the actual development of the crop, whereas SPAD-502 and GreenSeeker were more reliable sensors for predicting the biomass at harvest. Environmental stresses (e.g., heat-stress at flowering) during the growing period rendered the estimation of seed yield at harvest more difficult, as such tools were not able to take into consideration the external factors. This study also assumes that N requirements of Titicaca were low since the effect of increasing N fertilization treatments on seed yields and biomass at harvest was relatively low. This aspect was important from an environmental (for the ecosystems, e.g., eutrophication), social (for the public health, e.g., toxicity), and economical (for the farmers, e.g., by reducing costs from fertilizers) perspective. More frequent in-season observations with the POSs could substantially benefit the midseason management of fertilizer applications. In so doing, it will be possible to improve both the timing and the N fertilization rates, and therefore contribute to the enhancement of quinoa yields.

The yields obtained in this experiment, for the genotype Titicaca, were lower than those reported elsewhere in the Mediterranean region. Anticipating the sowing date (from May to April) is also envisioned to have a positive impact on the seed-filling phase by reducing the impact of water stress conditions occurring during summer months. Moreover, advancing the sowing date will consequently reduce the interference of abiotic stresses on the readings made by the POSs. In turn, this will result in a more reliable prediction of seed yields during the growing season. Hence, abiotic factors (heat-stress conditions and excessive N in the soil) can be considered as the main drawbacks of this research. For this reason, the present study strongly encourages further investigations in that direction, using reference plots (control) and advancing the sowing date.

**Author Contributions:** J.A.-B. (conceptualization, data curation, methodology, investigation, writing, reviewing, and editing), C.F. (investigation, formal analysis, writing, and reviewing), L.V. (data curation and investigation), S.T. (investigation), A.D.M. (conceptualization, supervision, and reviewing), S.O. (supervision and funding). All authors have read and agreed to the published version of the manuscript.

**Acknowledgments:** Thanks, are extended to the Istituto Tecnico Agrario for providing the experimental field. The authors wish to thank Euro Pannacci of the Department of Agricultural Food and Environmental Sciences of the University of Perugia for the provision of seeds.

**Conflicts of Interest:** The authors declare no conflict of interest.

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
