# Peer review of "Testing Proximal Optical Sensors on Quinoa Growth and Development"

_remotesensing, doi:10.3390/rs12121958_

Round 1
Reviewer 1 Report
Manuscript ID: remotesensing-801400
Type of manuscript: Article
Title: Testing different proximal optical sensors on the development and growth of quinoa
Dear authors,
Upon a careful reading and evaluation of your manuscript, I regret to inform you that I cannot recommend it for publication in the Remote Sensing journal. The main reason for this decision is the lack of novelty oriented for the remote sensing community. I believe that exploring proximal sensors to investigate the growth and development of plants with the investigated methods performed in the conducted experiment, is not suitable enough to a high impact journal as Remote Sensing since it does not bring new contributions to this community. However, the experiment is well conducted and the manuscript is presented accordingly. Still, I detect some deficiencies in its sections. In this regard, I have detailed my suggestions; which I believe will improve the overall quality of the manuscript content. I hope these comments are useful to you.
Does the introduction provide sufficient background and include all relevant references?
The introduction section comes with a short detainment on how proximal sensors, especially chlorophyll meters and reflectance sensors, have been implemented in the last decade to monitor N-uptake in plants and its consequences in growth and development. The authors focused on the type of sensor used and the evaluated data (mostly NDVI) to perform said task. In this regard, I advise the authors to provide more information regarding the methods implemented by these studies to analyze the data collected, as well as particular differences observed or discovered from one culture to another. Since the only novelty of this manuscript comes as an application orientation (as no study has performed this test in quinoa plants), the introduction section needs to support this information from the remote sensing point-of-view. Some questions to keep in mind: Did spectrally similar crops were already evaluated in this manner? What does your study differentiate from theirs? I believe that this section would be greatly improved if briefly introducing answers to such questions.
Is the research design appropriate?
I considered the research design appropriate. The authors did undertake the correct procedures to plan and performed the experiment correctly and collected data accordingly. However, the number of samples per field could offer a potential hindrance for the implementation of other operations and statistical techniques; and that is my main critique over this topic, which I’ll detail in my subsequent commentary.
Are the methods adequately described?
The method adopted to evaluate the collected data is the major flaw of this experiment. Maybe due to the experiment’s design, the authors were restricted to statistical methods like an ANOVA analysis and post-hoc Fisher tests to perform pairwise comparisons between samples and linear regression between the observed readings. As an example, their analysis ensured a coefficient of regression R2 = 0.71 (with a number of observations = 5) between the SPAD PO and Biomass. There is not a proposal of a new approach nor an implementation of state-of-the-art techniques to ensure better accuracy in their measurements. I suggest the authors evaluate new methods, currently being implemented in this type of research, and try to improve accuracy and find better predictions for the remaining crop parameters investigated here.
Are the results clearly presented?
Indeed, the results section is well partitioned and Figures and Tables are comprehensively enough. Still, since most Figures adopted a Black/White pallet, it became difficult to differentiate lines in most graphics even when giving different shapes (like triangles squares, hexagons, etc.). Some Figures (like Fig. 2) are hard to understand without the legends. Please considerer improving them. Sections like 3.1 and 3.2 are descriptive information regarding the collected data and Section 3.3 introduces some results from the remote sensing point-of-view. This is done for a brief moment by showing the relationship between Biomass values and POS measurements. The rest of it is composed of a direct comparison between POs measurements against each other. As previously stated, I believe that the authors should reevaluate their data and improve their method to achieve more interesting results.
Are the conclusions supported by the results?
The conclusions are partially supported by the results mainly because the authors did not propose a highly/solid research question, to begin with. Most of the concluding remarks are focused on the limitation of the conducted experiment. I suggest the authors remove the limitations detailed in the Conclusion section and promote general ideas of what their experiment was capable of achieving. These limitations are better suited to the Discussion section. There the authors should be able to even counterpoint most of its flaws.
In a general sense of this manuscript, the English language needs to be improved; there are some grammar and sentence structure errors in the text. I advise an inspection by a native speaker or a very careful examination in a subsequent read.
Author Response
Dear reviewer,
First, thank you for improving the quality of the manuscript. As regarding your comments:
-Introduction: this section has been improved by comparing with other crops (cereals and vegetables). The present study it's unique because up until now no studies had examined POSs on quinoa. Some of the suggested questions have been addressed in the introduction.
-Materials and Methods: a figure (Figure 1) and a table (Table 2) were included to enhance the description of our experiment. Regarding the number of observations, instead of putting the average per treatment we have included all the observations per plot (Figure 6, results).
-Results: the polynomial regression has been used rather than the linear regression (Figure 6). As requested, the quality of all figures has been improved, including the use of colors. Finally, the results have been reorganized in order to follow a more coherent order.
-Conclusions: the limitations have been removed from the conclusions.
Finally, the manuscript was revised by a native speaker.
Sincerely,
Jorge Alvar-Beltrán (corresponding author)
Reviewer 2 Report
Please see the attached.

Author Response
Dear Reviewer,
We would like to thank you for the support given to make this manuscript much better. Here below some of the responses addressing your comments:
Abstract
-The title has been modified accordingly to: Testing proximal optical sensors on quinoa growth and development
-(R2 0.71 and 0.65) has been modified to: coefficient of determination, (R2 = 0.71 and 0.65, respectively), and statistical significance (p ≤ 0.05)
-As requested, key words have been modified to: Nitrogen fertilization; Normalized Difference Vegetation Index; Yield and biomass prediction; Abiotic stresses
Introduction
-A reference to support the statement on the nutritional properties of quinoa has been added.
-As requested, the following sentence has been added to the objectives of this study: Any advances on testing POSs on quinoa development and growth as well as on the ability of grain yield in-season prediction will benefit farmers on improving fertilization decision-making.
Materials and methods
-The sub-chapter 2.3 has been modified to: 3 Proximal Optical Sensing
-The equation for the coefficient of determination has been added to the manuscript.
-Overall, the materials and methods section has been boosted up, with new figure and a table (Figure 1 and Table 1)
Results
-The section on agrometeorological conditions has not been moved to Materials and Methods as a previous reviewer suggested to leave it as results. We have now modified the name to 1. Soil and Weather Conditions. Some of the results provided in 3.1 are essential for the discussion section.
-L171-174. The “x” and “y”-axes for this legend were displaced in the manuscript forwarded to the reviewers. This has been amended accordingly throughout the text.
-L180 has been improved accordingly.
-As requested, section 3.3 has now been moved to 3.2, and vice versa.
-The way R2 is written has been revised throughout the text
-All observed values for Figure 7 (now Figure 6) have been included in the graph.
Discussion and Conclusions
-As requested we have improved both sections, besides of including information about the drawbacks of our research and potential beneficiaries.
Thank you for your inputs,
Jorge Alvar-Beltrán (corresponding author).
Reviewer 3 Report
Dear authors, my comments in the attached pdf
The most important notes:
1. if control (0 kg N) yield is higher than the yield of fertilised plants this probably means that the experiment was also influenced by other factors - such as soil variability, high soil nutrient content (including N) which was influenced by e.g. forecrops - I suggest checking this carefully, or repeat the experience this year. In the current text - the discussion and conclusions are, in my opinion, too optimistic and biased in relation to the obtained results.
2. Figures require better development - please consider introducing color, better resolution and better description and construction of the legend
3. statistical relationships between spectral measurements (SPAD and GS) and simple image interpretation (Canopeo) should not be compared.

Author Response
Dear Reviewer,
We much appreciate and agree with the comments and revisions provided. Here below our responses to your comments:
- Listing of co-authors has been amended accordingly.
- Keywords have been modified accordingly.
- The Latin name of quinoa has been added to the manuscript.
- The BBCH scale has been incorporated into the manuscript and has been modified accordingly in the results section part.
- L103-104. The cumulative growing degree days (CGDD) equation has been added to the manuscript. In addition, for the section analyzing (L195-L201) the different phenological phases, CGDD has been added after DAS.
- Figure 1. When converting to a PDF the units disappear. They are now included
- CGDD has been added after DAS
- L141-144. The title of sub-chapter has been modified to 3 Soil and Weather Conditions
- Figure 2. When converting to a PDF the units disappear. They are now included
- Figure 3. As suggested, we have decided to delete this figure as it was to some extent confusing. Other reviewers recommended to do the same.
- Discussion: as requested we have added a sentence on the discussion section explaining why N fertilization did not have an impact on final yields.
Thank you very much,
Jorge Alvar-Beltrán (corresponding author)
Reviewer 4 Report
Comments about the Paper: “Testing different proximal optical sensors on the development and growth of quinoa”
The paper deals with the investigation of three proximal optical sensors in order to monitoring crop development and nitrogen status of plants. The results are promising for a plot level for the case study area. In overall, the work is interesting and I believe the topic of this paper is appropriate to be published in Remote Sensing.
All in all my recommendation is to accept the paper for publication, subject to minor revision.
Please find in the attached file the amendments which I believe are required prior to accepting the paper.

Author Response
Dear reviewer,
We would like to thank you for the useful comments and for substantially contributing to make this paper better. All the comments have been addressed as follows:
- A figure with the area of study and experimental design has been included in the paper (Fig.1).
- As requested, a flowchart of agronomic activities, parameters measured, and number of samples has been added into the document (Table 1).
- A sentence about the calibration has been provided in section 2.3 Optical Sensors
- As requested, the lineal regression, coefficient of determination and significance are now reported in a Table (Table 4).
- A table has been added to the discussion with the observed pros and cons of each device (Table 5).
Best,
Jorge Alvar-Beltrán (corresponding author).
Round 2
Reviewer 1 Report
Dear authors,
I've verified your new submission and, upon a second reading, I'm reconsidering the recommendation of your manuscript. Although you stated in your response letter that this study is "unique", it is still not a novelty paper in the methodological view from the remote sensing community. Please keep this in mind. Regardless, the corrections and improvements conducted in the revision phase were correctly performed. I believe that, in the conditions promoted by this special issue, the interest and information divulgated here are adequate to its readers. I hope that the advice and recommendations provided will be carried in future work. I wish luck to you and your team.
Best regards.
Reviewer 2 Report
Please see the attached

This manuscript is a resubmission of an earlier submission. The following is a list of the peer review reports and author responses from that submission.
Round 1
Reviewer 1 Report
The research might sound promising, especially for growers that depend on a budget, looking for affordable sensors. However, although the conclusions are carefully formulated, the results can hardly sustain them.
This comes from an experimental design that should be better planned. Specifically, the data obtained only after one year of cropping is insufficient, especially with the climatic coditions depicted (high temperatures in the second half of the cultivation period).
Moreover, under such conditions, the authors claim that sensor values can predict biomass yield, but no mathematucal model is proposed. Furthermore, in specific cases, the standard deviation for biomass values is very high, up to 50 % of the mean, under 100% treatments. This indicates high variability in measurements.
For imaging, at least one-two pictures should be provided, maybe under supplemental material.
Reliable results should be obtained at least after two crops.
Mathematical modelling should be included.
Reviewer 2 Report
I have reviewed the manuscript remotesensing- 708620 and I think the authors have carried out an interesting study to relate ground-based remote sensor measures with quinoa crop development under different nitrogen status.
Nevertheless, I consider this article is not suitable for publication in Remote Sensing journal. This manuscript presents a very basic level of analysis with sensorization and does not include any novelty in the field of remote sensing (neither the sensors nor the treatment of the information). This is the biggest weakness of this work. The authors only used the direct measurements of the different sensors and compared them with the data collected in the field. Therefore, I suggest that this work should be sent to a journal more appropriate to the type of study (e.g. Agronomy mdpi).
I consider that there are some details that should be revised in order to improve the manuscript. My comments are detailed below:
First of all, the template used in this manuscript is not the one used in Remote Sensing Journal. In fact, it is the template of Land Journal. The authors should be more careful with the presentation of the manuscript to avoid any confusion.
Additionally, there are some flaws in the article format. Authors do not follow the rules of this journal. The authors should revise the references throughout the manuscript. They are not indicated as the guide for authors suggests. The same with the reference section. The references are is another style than mdpi suggests. Revise the name of the sub-sections (section 2). All of them are 2.1.
The keywords can be improved. For example, NDVI should be included. Is Italy so important to be included in the keywords? The Mediterranean climate (included in the title) is a more global idea.
In the introduction, there is no information about the importance of the quinoa crop. It is named as a superfood in the keywords, but the authors do not put it in value in the introduction.
L 81. Which is the Datum of the coordinates? This information should be more accurate.
L 104-115. The authors showed little information about remote sensor measurements (section 2.3). More details about the data measured and the collection conditions are required.
L 119-122. The authors suggest different N-fertilization levels and gave them a very long name. These N-fertilization levels should be renamed with a shorter name in order to improve the reading of the manuscript. Of course, the first time they are rename, a more extensive explanation should be showed. This shortening of the name of the N-fertilization levels should reduce the text and the figure captions.
L 126-129 and Table 1. Are that data results? I consider this information should be included in the 1 Experimental site and treatment details section.
Figure 1. Authors should label the X-axis.
Figures 2-7. The legends should be included in the figure captions and should be reduced when possible. L 172. Is this legend necessary in this figure?
Figure 3. I do not consider the use of an accumulative percentage as the best way to compare the different phenological phases. It is difficult to observe the numerical information and to compare.
Table 2 and Figure 6. The information of Table 2 can be included in Figure 6.
Reviewer 3 Report
Apparently, the manuscript was formatted for MDPI Land, not for Remote Sensing. Not surprisingly, the manuscript did not contribute to the remote sensing community. The authors basically used three green ground-based sensors for monitoring plant biomass under different nitrogen treatments nad evaluated the performance of the sensors at different growth stages. The authors mistakenly claim these as remote sensors, while they are proximal sensors as mentioned in the introduction section. The manuscript may better fit for agronomy related journals. So here I can not suggest the manuscript for Remote Sensing.
Reviewer 4 Report
I suggest the following minor changes and additions to the authors:
numbering of subsections in Chapter 2: please change to consistent numbering: 2 .; 2.2; ... description of experimental fields from chapter 3 should be moved to chapter 2 (2.1) description of weather conditions - as above (2.2) Figures 2, 4, 5 - interpretation of letters from the vertical column is illegible - I suggest to unify the description:Letter “ a”: mean values that are significant at 5% probability level using Fisher test”
Letter “b”: …
Reviewer 5 Report
What is proposed looks like too much a summary of technical reports. First of all, this manuscript didn't follow style guidelines for this journal.
#1
2.1. Experimental site and treatment details
You said GBRSs are useful tools for monitoring the development of crops and nitrogen status of plants. Why didn't you measure N content of leaves?
#2
2.1. Remote sensor measurements
The outputs from GreenSeeker and Canopeo App are also influenced by the background and the structure of plants. Therefore, I have serious concerns over the use of statistics.
#3
I think that the authors should investigate much more comprehensively the possible explanations of the links between nitrogen status of plants and the reflectance used in the three devices.